# “My Mother Persuaded Me to Have More Children”, Understanding the Influence of Social Network on Fertility Behavior in Sub-Saharan Africa

**DOI:** 10.3390/ijerph21040396

**Published:** 2024-03-24

**Authors:** Stephen Okechukwu Chukwudeh, Akpovire Oduaran

**Affiliations:** 1Department of Criminology and Security Studies, Faculty of Social Sciences, Federal University, Oye-Ekiti 371104, Nigeria; 2Community-Based Educational Research (COMBER) Entity, North-West University, Potchefstroom 2531, South Africa; akpovire.oduaran@gmail.com

**Keywords:** children, fertility, mothers, population growth, sub-Saharan Africa

## Abstract

**Introduction:** Despite the adoption of an antenatal demographic transition model in sub-Saharan Africa, population growth continues to soar in the region. The reasons for population growth are nebulous and should be approached from different perspectives. Inadequate attention has been paid to how social pressures shape reproductive behavior. Thus, this study examines the association between social networks and fertility behavior in sub-Saharan Africa. **Methods:** This study used a cross-sectional design that employed a qualitative method to conduct 28 in-depth interviews among married women. Data was collected in 2023. Thematic analyses were utilized to interpret the results. **Results:** Parental pressure, the desire for more children, the desire for male children, values placed on children, norms, and pronatalist culture were associated with high fertility in sub-Saharan Africa. **Conclusions:** Thus, there is a need for more awareness of family planning and contraceptive use in order to reduce fertility in sub-Saharan Africa.

## 1. Introduction

Several governments, non-governmental organizations (NGOs), and relevant stakeholders have made concerted efforts to understand the fertility dynamics in Africa. This was the reason for the seminar on reproductive change in sub-Saharan Africa (SSA) that was jointly organized by the International Union for the Scientific Study of Population (IUSSP) committee on fertility and family planning and the Africa Population Policy Research Center (APRC) in 1998. The seminar suggested the following as reasons for the fertility boom in the region: female age at marriage, poor contraceptive use, urbanization, and illiteracy. However, these postulations are not exhaustive, as negligible attention has been paid to social networks as a strong determinant of fertility behavior in sub-Saharan Africa. This is not surprising, as African governments have long adopted the pronatalist demographic transition model of Europe that has consistently failed in the region.

Africa accommodates about 20% of the world’s population, and [1] estimated that sub-Saharan Africa will constitute at least one-third of the projected African population by the year 2050. Large population sizes have implications for demographic dividends. However, the dividend could be negative in sub-Saharan Africa, where basic social amenities are relatively scarce. This accounts for the established policies on the use of contraception, family planning, and restrictive abortion policies to reduce the rate of fertility. These formal policies have not been totally effective in sub-Saharan Africa. The narrow efforts toward mobilizing those in the health sector to address implementation challenges targeted at fertility reduction have ignored the implementation of fertility control through social network mechanisms.

Consistent with the above, Refs. [2,3] opined that individuals’ fertility behavior relies absolutely on the fertility behaviors of their social networks. What are the direct mechanisms through which social networks influence fertility behavior at the individual level and what are the implications of these routes to high fertility in sub-Saharan Africa? Social network is an ongoing system of social relation, interaction, and sociation that is based on the value placed on children. This value has influence on fertility tempo and quantum. For instance, women do not just adopt family planning techniques or use contraceptives independently, without the consent of their socially accepted network of peers, like husband, friends, neighbors, in-laws, and family members [4,5]. Further, in some societies in Africa, the fertility behavior of women is influenced by the fertility outcomes of other women within the same clan or society [6,7].

It should be noted that fertility intention, the value associated with children, the support received by biological parents, and the passion attached to having a large family size begin and are sustained through social interactions [8]. Social relationships and interactions influence social acts such as childbearing [9]. Contextually, as used in this article, the term social network does not refer to the use of social media for interaction but to the social relations that shape beliefs and norms. While beliefs and societal norms are abstractions, it is the sociation within the society that propels people to adhere to human abstractions such as beliefs, thus forming a social structure that endures in the society for many years. This study focused on how social network influences fertility in sub-Saharan Africa.

The social network is an important source of information and interaction in the contemporary world, especially as regards fertility behavior. There is high trust and reliance on one’s social network, such as family members, friends, relatives, schoolmates, colleagues, and others for vital information on the timing of birth, when to get married, whom to marry, the number of children to have, and the spacing of childbirth. Contemporarily, scholars have made little effort to understand the role of social networks on fertility outcomes in sub-Saharan Africa, especially among migrants. It is ideal to understand the process through which migrants adopt their fertility behavior in sub-Saharan Africa. Against this background, this study examines the relationship between social networks and fertility behavior in sub-Saharan Africa, using Nigeria as a case study.

## 2. Materials and Methods

### 2.1. Design

A cross-sectional design was employed for the study. This design was appropriate for collecting data from a large pool of the population and observing their behavior. The study utilized a qualitative method. This is ideal for exploring the interpretative understanding of the experience of married women in the study location. The study was conducted in Nigeria, a country where people from several countries in Africa can be found. Due to the dynamic nature of the country and the heterogeneity of the population from different tribes, languages, and people, the populace was used to represent sub-Saharan Africa. The city of Lagos in Nigeria is one of the most populous cities in sub-Saharan Africa, with a growing population. The city is the commercial hub of West Africa and was created in 1967, shortly after the independence of Nigeria. For this major reason, the study sample was selected from this city.

### 2.2. Study Participants and Sampling

The lead researcher first met a female counselor that specialized in family planning issues. By means of the counselor, the first respondent was met. The first counsellor, by means of referral, also introduced a second counsellor to the research team. The two counselors served as the key informants for the study. By means of referral and the snowball sampling technique, the researchers were able to approach the respondents. All the respondents gave oral consent after it was explained to them that the research was purely for academic purposes. There were two categories of people interviewed for the study. The first group of participants were the two key informant interviewees (KIIs), who were female and senior executive members in privately owned health facilities. The key informant was purposely selected from two health facilities (one key informant per health facility).

The second group of people were twenty-six respondents for the in-depth interview (IDI). The respondents were selected via referral and snowballing. These respondents were interviewed via face-to-face interview. These respondents were approached as they exited the hospital. Although respondents were approached via snowballing and referral, approximately five people still refused to participate in the study. This was because of their busy schedules. The duration of the meeting was between 40 and 60 min, and there was no return visit for the interview. The interviews were conducted at the convenience of the respondents. Participants were recruited until saturation, when there was no new theme observed. The inclusion criteria were: being a married migrant woman, being 18 years of age or older, living with husband, having children, and being willing to participate in the study. All other factors were the exclusion criteria, including being a single mother.

### 2.3. Interview Guides

A semi-structured interview guide was used for the study. This was sacrosanct to enable respondents to freely express themselves. The issues covered by the guide included: time of entry into marriage, age at first birth, planned number of children, actual number of children in the family, influence of biological parents on fertility outcomes, influence of grandparents on fertility, influence of significant others on women’s fertility behavior, family reproductive history and gender preference, and method of pressure from their network. The interview guide was piloted in a public health facility. In approaching the respondents, the researchers first teased them with other issues in order to make them relax and build a relationship with the women before engaging them on the research topic. This was very important due to the volatile state of the sub-region.

The research was conducted between August and December 2023. The length of time was due to the respondents’ convenience and the appointment schedule between the respondents and the researchers. An audio tape recorder was used to capture all comments so that nothing was missing from the analyses. Field notes were used to jot down points after requesting permission from the respondents. The themes that formed the objectives of the study were derived from the responses of the respondents. The total duration of each interview ranged from 40 min to 60 min and the interview ended after data saturation. To efficiently solicit information from the respondents, the researchers first introduced themselves. After the introduction, the researchers began by discussing the current economic situation and how families survive amidst growing inflation. After common ground had been gained during the engagement, the following questions were posed before the introduction of the main research questions: How do you cope with the current economy? Do you plan on having more children? etc.

### 2.4. Data Collection

Two female research assistants were recruited for the data collection. Only female research assistants were recruited for the data collection due to the sensitive nature of the study. The presence of the married female research assistant also prompted the female respondents to be free to open up in their responses. The lead researchers trained the research assistants for one week on the rudiments of data collection within the context of the study location. The KII and IDI were conducted in a private location, as specified by the respondents, and the only persons available on each occasion were the respondent and at least two members of the research team. This was to ensure privacy, minimize distraction, and give the respondents the freedom to express themselves without barriers.

By means of referral and snowballing, the respondents were met, while the key informants were met by means of purposive sampling and referral. The interview guide for the study had been pretested earlier, and corrections were made where necessary to capture the objective of the study. Two female research assistants and the lead researcher were scheduled at each session to conduct the interview with female respondents during the IDI. All the interviews were conducted once with the respondents, and there was no repeat visit so as not to endanger the respondents. The duration for each session of the IDI was between 40 min and 60 min. All interviews were audio recorded and conducted in the English language. Data collection and transcription took 6 months.

### 2.5. Data Analyses

The thematic analysis approach, which requires data generated from the field to be analyzed qualitatively in order to generate themes and sub-themes, was adopted in the study. The themes were generated from the experiences of respondents. Thus, the researchers were able to compare and contrast data to efficiently describe the relationships between themes. The two researchers were involved in the coding using Microsoft Word. An initial line-by-line open coding was used to categorize and summarize the data, followed by more focused coding centered on the significant codes and categories that tended to be more conceptual. This facilitated comparative analysis, aimed at comparing data to data and data to codes and categories in order to identify similarities and differences. This method has been used effectively in qualitative reports [10,11]. The method entails generating codes from data, and when a group of codes is repeated in a particular way, it becomes a theme. Themes were developed by the compare and contrast approach, which entailed a line-by-line analysis of words, sentences, and content. Each line was read, and contextual meaning was deduced. This was followed by sentence-to-word, and sentence-to-sentence analysis to know how one word and sentence differed from the preceding word and sentence and the meaning therein. Text pairs were compared to determine their contextual meaning among the respondents. This method strengthened the data.

## 3. Results

The findings of the study were presented in the context of three objectives, which focused on pressure from biological parents, the grandparent effect, and adherence to the norms/culture of the society. These objectives aid in the interpretative understanding of the fertility behavior of mothers. The major themes in the study were derived from the data.

### 3.1. Pressure from Biological Parents

We inquired from childrearing parents who had more than four children what motivated their choice of their actual family size. A respondent reported:

My biological parents persuaded us to have more children. When I got married, my husband and I planned to have only two children but with the severe pressure from my parents, especially my mother, we now have five children. (P1)

Another respondent noted: 

My parent does not believe in having a few children. They believed it was Western European way of life. In fact, my father accused my husband of allowing what he watches in the television set to influence his household. We experienced serious confrontation until we had more children. (P2)

This highlights the role play by parents on their children’s fertility behavior. A few stressed that the practice of having a few children is weird within the context of the way of life in Africa. Another respondent narrated her experience: 

My parents stopped communicating with us even though at that time we relied on them for moral and financial support. They would not pick our calls, neither would they call us over the phone. Our crime was our decision to have only two children. There was a day my parent arranged for a meeting with all the children from my extended family and the issue that was discussed was the number of children I had. The pressure made me to have more kids. At the moment, I have five children and I do not know when I would stop having more children. (P3)

These narratives highlight the experiences of women with a relatively large family size. The challenges within the family could be severe, and if not properly managed, they could result in a failed marriage. A respondent noted: 

I currently have six children for my present husband and two children for my late husband. So, I have a total of eight children. Initially, when I married my current husband, I planned to have just two additional children for him, but his parents used that to taunt me every time we had a discussion. I was threatened to have more children if I want to sustain my marriage. That made me consent to having more children. (P4)

The importance of large family size cannot be over-emphasized in the context of living in Africa. Large family size is also perceived as a source of security within the community. For instance, a respondent stated: 

My mother persuaded me to have at least five children in order to have a family. In her opinion, a typical Africa family is large, and this serve as a security for them. People in the rural community are scared of large family as nobody will dare trespass into their property. Therefore, she concludes that I should endeavor to have a large family. (P5)

Contextually, the size of a family and the number of children that a woman has is often used to assess her value to the family and community. The larger the family size, the more value is attributed to the woman. Large family size is seen as a form of security and an Africa lifestyle.

### 3.2. Grandparents’ Effect

It was also observed that aside from the pressure from biological parents, grandparent also influence the size of a nuclear family. For instance, a respondent reported:

I grew up in my grandmother household. I love, respect, and listen to her advice because she trained me, and I have live almost all my life with her. When she visited me after I gave birth to my second child, she lamented that she was expecting a twin from me because her own children had always produced a twin. When she learnt that my husband only wanted two children, she grew angry and began mounting consistent pressure on my husband for more children. This propel us to have five children now. (P6)

Another respondent noted: 

My grandmother loves children. She always comes to pick my children immediately they reach five years old. At the moment almost all my children are living with her. This makes child rearing easy and flexible for me. Only one out of my five children are living with me. This is because my grandmother insisted that the other four children should live with her. She also pressure us to have more children if we wants her blessings. (P7)

The influence of grandparents on the number of children being born by parents in Africa is massive due to cultural undertones. Culture warrants that parents to listen to and take advice from their parents, which includes the number of children they give birth to. For instance, a respondent noted:

My grandmother appealed to my husband and myself to have more children because our three children are now in the university. She explains that it was not ideal for us to live in our house without noise from children. She argued that it was not the culture of Africa and thus, it has a negative implications. That propel use to have additional two children. (P8)

Another respondent noted:

My grandmother lives with us in our house. Her constant tears and threat to kill herself if we don’t give her more children influenced my husband to appeal to me for cooperation to have more children. (P9)

The above comment aligns with below made by another respondent: 

“How would my friends have more grandchildren more than me. It is an insult”. These were the words often spoken by my mother in law to put severe pressure on my husband to have more children for her. Due to the extreme love my husband have for her mother, and in order to satisfy her mothers need for children, my husband often persuade me all night to get pregnant and give him more children. The pressure is much for me. Thus, I had to comply to save my matrimonial home. (P10)

These examples show the intricate influence of social networks on reproductive and fertility behavior in Africa. Contextually, parents crave the blessing of their own biological parents and grandparents. This conversely makes them listen to them on issues related to fertility outcomes. 

### 3.3. Norms and Culture of the Society 

Aside from parental influence on the number of children that people have, other factors that influence childbirth are societal norms and culture and the value attached to the gender of the children. A respondent reported:

After I had my third child, my husband was very furious with me because all my children were female. He complained that his friends do label him as lazy because all his children were female. For that reason, we decided to continue reproduction in our quest to get a son. (P11)

This shows the pain people with only female children experience in their everyday lives. Another respondent noted: 

At the moment I have four children but unfortunately they are all female. This is a cause for worry as my husband is not happy with the situation. He complains that there is no male to carry on his legacy and he crave for a male child. I am ready to make him happy, so I am going to try again to see if I can get a male child. (P12)

These examples show the importance people attached to male gender, and this influences the number of children within the home. In addition, the number of children a woman gives birth to has cultural undertones. For instance, a respondent reported:

It is not the culture of Africa for a family to have only a child. It is our tradition to have large family size. There is always pressure from parents and family members to sustain existing culture even as regard to reproductive behavior. (P13)

This idea was supported by a respondent who said:

Why would I have just a child or two when my mother had eight children? It is not our value system here in this country. As for me, I would have more children. I love the serenity of having several children around me. (P14)

Another respondent noted that:

Having large family size is an intergenerational practice. Why would someone desire small family which is contrary to the norms of the society? For me I will continue to give birth until my body is tired. I cannot use contraceptive or indulge in family planning. (P15)

These examples highlight the influence of societal norms and culture on the reproductive behavior of people.

## 4. Discussion

This study observed that the pressures and demands from biological parents for more children or grandchildren were one of the factors responsible for large family size. While the mother may have a met need for children, the husband may still need more children in the family. This shows that within the family, there is a disaggregate number of children desired by each spouse. This is in conflict with the National Policies on Population for Development, formulated in 1988 and amended in 2004, which have the goal of reducing population growth in Nigeria. Cultural values for children will continue to propel parents to have more children, which is contrary to the global goal of reducing population growth. This is propelled by means of parental social networks that pressure spouses and children to have more children. Nigerian fertility behavior is pronatalist, despite anti-natalist population policies in the country. This shows that population policies in Africa are not in tandem with the reproductive reality on the continent. This is not surprising, as previous studies have reported low contraceptive use and family planning upkeep among married women in sub-Saharan Africa [12,13]. 

As long as women continue to experience pressure from their spouses and the society at large to have more children, nuclear families will continue to have large family sizes. This does not exclude women in polygamous families. Women with only a few children are often abused and deprived of their rights within the family. This report aligns with previous studies, such as [14], which stated that social pressures from friends, colleagues, and family members were associated with the timing of marriage and fertility of women in Nigeria. Also, Ref. [15] concludes that cultural norms, as shaped by social networks, influence fertility behavior. Other studies have reported that religion and negative perceptions towards family planning and contraceptive use were responsible for the high fertility outcomes in Nigeria and Africa [16,17]. Although both men and women are responsible for the reproductive cycle, contextually, society puts more pressure on women when it comes to childbirth. 

The nonuse of contraception among women who are still sexually active makes them susceptible to pregnancy. Many women who find themselves pregnant do not abort due to perceived illegality in Nigeria [18]. The situation is made worse by a short birth interval that does not correspond to the recommendations of the World Health Organization [19], which stipulates 24 months after birth before the next conception. This study observes that short birth intervals favor the spouse who has a passion for large family size within a short period of time. However, families that desire fewer children will have a contrary view, as they are likely to delay conception. This practice is strongly entrenched in the cultural values attached to children.

Previous studies [20,21,22] posit that there is a relationship between large family size and the intergenerational transfer of wealth from one generation to another within the context of Africa. They argued that it is not economically rational to have a limited number of children. This is because having fewer children decreases the net wealth and income of a family and increases the liability. The reason for having a large family size in traditional African societies was because children were sources of wealth and social prestige. Therefore, parents had many children because they saw children as a source of financial and social support when they were older. However, within the context of Nigeria, [23] stressed that large family size causes a huge economic burden, and thus, practical efforts should be undertaken to reduce the persistently high fertility in the country. Nonetheless, the study found that, despite development over the years and government policies to reduce family size through the practice of contraception, in reality, the people still hold firm to traditional norms for large family sizes.

### Limitation

This study was conducted among migrants who reside in Lagos, Southwest Nigeria. A replication of the study in another location may yield different results. This is a qualitative study, and the results are purely the opinions of the respondents. However, this study enables the interpretation of respondents’ responses to attain a better understanding of their experiences.

## 5. Conclusions

Our study found that pressure from biological parents and grandparents, values associated with having children, especially the male gender, and norms and culture of the society were implicated in the high fertility rates in sub-Saharan Africa. These social and cultural practices in a patrilineal society influence fertility behavior in the region. Social networks shape the values attached to beliefs and norms that are practiced in society. This invariably implies that social networks influence fertility behavior. This has implications for population growth and sustainable development in sub-Saharan Africa. 

## Data Availability

Further data would be available where needed.

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
