# Peer review of "“My Mother Persuaded Me to Have More Children”, Understanding the Influence of Social Network on Fertility Behavior in Sub-Saharan Africa"

_ijerph, 2024, doi:10.3390/ijerph21040396_

Round 1

Reviewer 1 Report

Comments and Suggestions for Authors

There is interesting information in this article but I'm not sure the framing is correct. I would have expected more discussion of networks. As it is this article is really more about social norms. I think it would be stronger if it were focused on social norms, how they are enforced (which does include networks) and passed on. 

The conclusion in the abstract and later on- that more awareness of FP and contraceptive use is too general to be useful and not really tied in any specific way to social norms or networks. I suggest rethinking what the implications are of your results - simply increasing FP knowledge, for example, is unlikely to address the barriers you identify in the results.

Please check the definition of social networks on line 46, "an ongoing system.... based on the value of children". This does not seem correct to me, perhaps an English language issue.

Study participants and sampling- I think the process is appropriate, but it is difficult to understand as explained. I think a heavy edit is needed of this section. In paragraph two under Study Participants (line 92), I do not understand how the respondents were selected.

Interview Guides. (p. 100) - from the text it seems that no network questions were included (who you are connected to, who do you rely on for information, resources, etc.). This make me suspect this study is more focused on social norms than networks. Please address by further explaining the methodology or shifting the framing of the article.

Results: There is convincing evidence in the result section on the influence of norms on FP use. However, no information on networks specifically. This information is not really novel, but perhaps builds evidence on the influence of norms on FP. I think a more interesting framing of your results would be on norms, and how they are enforced via sanctions. Do individuals challenge/violate these norms? If so, how, what makes this agency possible? 

Discussion: the flow and logic of the discussion is not strong, and I don't find it well linked to the study results.

Limitations. I did not realize until I reached the limitations section that this study was conducted among migrants. That is hugely important and should be included in the framing of the paper and throughout. There is scholarship on norms and networks among migrants and how they may or may not differ from other populations. I think that your paper should address and weave that theme in throughout.

Comments on the Quality of English Language

A thorough editing of the English is needed. There are grammatical errors, and in addition there are areas which are difficult to understand because of the language challenges.

Author Response

Thank you, I appreciate the proofreading and comments of reviewer 1. The article is on social network and not on social norm. This terminology look different but it is different to sociologist, demographer, and social psychologist. The context of social network influences people's behaviour and action to obey or refute social norms. social norm is an abstraction which can only be enforced by sociation or social network. sociation or social network is a widely use term in sociology (see Bernardi and Klarner, 2014; Keim, Klarner, and BAernardi, 2011 etc). Please note that the term social network is not the general usage of online network of interaction between people. This has been clarify in the introduction.

The English language in the text has been improved upon. The sampling has been clarify to show how sampled respondents were recruited.

The context of network has been embedded in the interview guide as questions were curated from previous study on social network

The social norms as stated by reviewer 1 is reinforce by the network of people and not by the object itself. this is made possible through network sociation. For instance, the Yoruba tribe in South-west Nigeria prostrate to exchange greetings or pleasantry. In other region within the same country, this singular act is not a norm among the male. It is the social network that determines and influences such action. This study is on how social network of people influences the reproductive behaviour of people.

The discussion section has been link appropriately with the result.

Thank you so much for your valuable, insightful and engaging comments.

Reviewer 2 Report

Comments and Suggestions for Authors

The introduction

The authors do not adequately introduce the subject matter of the study, which is the influences of social networks on childbearing in Africa. There is no description of studies that have conducted research on the subject to justify the research question. It is possible that it is a topic that has been studied in previous studies that have not been reported.

Method

The description of the methodological design is very deficient, it only specifies that it is a descriptive cross-sectional study but not the type of study conducted within the range of descriptive studies.

Study participants and sampling

The description of the sample of participants is very confusing, lacking tables that could better clarify how the selection was made and the number and sex of the participants. The same applies to data collection and resutls

The discussion lacks a good literature review that can compare the data

Author Response

The introduction section of the study has been enriched to capture the concern of reviewer 2.

Method: The methodology has been enrich to capture the migrants that lives in Lagos and how their network of sociation influence their reproductive decision and behaviour.

Study participants: In a qualitative study, it is not compulsory to use table when specifying the participants. The authors are comfortable with the current usage. The authors have explained the contents appropriately

Discussion: Content and context of study differs and that is the beauty of qualitative study that reinforces the interpretative understanding of a particular subject.

Reviewer 3 Report

Comments and Suggestions for Authors

Congratulations on the study. List these considerations to help in the writing process:

ABSTRACT: In "Method" add the guiding question or variables addressed in the research. Add the period in which the data was collected. Include how the data was analyzed.

INTRODUCTION: LINE 35: Describe more data on the negative impacts of population growth.

INTRODUCTION: It is important to conceptualize "social network", as this is what the authors will be analyzing. Conceptualizing "social network" will help us understand how the authors constructed the questionnaire.

INTRODUCTION: The study also considered the online social network?

INTRODUCTION: what is the reference used for the statement written in lines 55, 56 and 57?

METHODS: make explicit the inclusion and exclusion criteria for the participants. What characteristics were expected of the participants.

METHODS: How was the research question posed? How was the study's triggering question posed?

METHODS: A theoretical framework was used to delimit what the authors understand as a social network. This directly affects how the questions are constructed.

METHODS: LINE 154: "This method made the data analyses robust." - I think this sentence can be avoided. The term "robust analysis" has a specific meaning in quantitative studies.

RESULTS: It is important to create a table to characterize the participants. It was listed in the methods that different variables would be addressed, but they do not appear clearly in the results. For example, characterizing income and age could help interpret the data. Would people who were older or younger, and with lower or higher incomes, have the same weight in terms of the influence of the social network?

RESULTS: Psychological and financial impacts of deciding to have fewer children.

RESULTS: It is important to highlight the differences in how the wife was pressured compared to how the husband was pressured to have more children. Did women suffer more direct and incisive pressure when compared to men?

RESULTS: Doesn't public birth control policy provide for actions to promote population security, to reduce the need for the family to be large as a support for family security?

RESULTS: To explore what lies behind the devaluation of female children.

RESULTS: LINE 266: The participant in the study reported not being able to use contraceptive methods or carry out family planning. Are there religious factors behind this prohibition? What part of the culture highlights these prohibitions?

DISCUSSION: I suggest pointing out which ideas that religion reinforces encourage greater numbers of children.

DISCUSSION: I suggest including the prevalence rates of sexually transmitted infections in the discussion. Since the participants have a profile of not using condoms during sexual relations.

DISCUSSION: The authors could carry out a dialog between the results of this study and the strategies adopted by public birth control policies in sub-Saharan Africa. Checking the assertiveness of the actions taken by public policies.

REFERENCES: I suggest reducing the use of the proportion of articles with more than 5 years of publication.

Author Response

The variable address and analyses method has been included in the abstract, while the duration of data collection is included in the method section of the main work. 

social network has been conceptualised as requested.

The inclusion criteria has been added as married women with children.

line 154 "robust analyses" has been deleted

While thanking the reviewer for his/her valuable comments, it must be stated clearly that prevalent rate, religious influence, strategies adopted for birth control, reasons behind devaluation of female children are beyond the scope of this work. However, these may be consider in future work

Thank you for your valuable contributions. This is highly appreciated

Round 2

Reviewer 2 Report

Comments and Suggestions for Authors

- The introduction has improved a lot but there is a new content introduced that is not well understood, it is the one that refers to migrants, what kind of migrants does it refer to? Could you explain this paragraph better?

- Important gaps in design and methodology remain:  What type of qualitative design have you used?What was done to counterbalance the presence of the researcher?

Did the researchers discuss their own behaviours and experiences in relation to the experiences of the informants?

Have other methods (triangulation) been used in data collection to determine the congruence of results?

Did the researcher discuss their interpretations with other researchers?

Author Response

Thank you for your valuable contributions to enrich the quality of our manuscript.

  1. The qualitative study is ethnographic
  2. only female research assistants were present to make the respondents comfortable to express themselves. Respondents were also informed that this was purely for academic purpose
  3. All report in this article are sorely that of the respondents
  4. Data triangulation was used to determine the congruence of result
  5. yes, results and interpretations was discussed with other researchers
  6. Data was collected among women who live in other country (migrant)
  7. Thank you once again for your contributions to enhance the quality of our manuscript

Reviewer 3 Report

Comments and Suggestions for Authors

I congratulate the authors for the adjustments made to the manuscript.  I would like to point out a few outstanding items:

ABSTRACT: Add the period in which the data was collected [Include at least the "year" in which the data collection took place].

METHODS: Explain the inclusion and exclusion criteria for the participants. The authors explain how the process of inviting participants and data saturation took place, but they don't make it clear what the inclusion/exclusion criteria were for participants, which people could be invited to the research or not. [Example - the inclusion criteria were: being a woman, being over 18, working in a certain profession.... And the exclusion criteria were: not responding to the third contact attempt...]

METHODS: How was the research question posed? How was the question that triggered the study posed? [Please describe explicitly how the main open question of the research was written, so that other researchers can replicate the methodology of this study in the future].

Author Response

Thank you for your valuable comments to improve the quality of this manuscript.

  1. Year of data collection has been included in the abstract section as requested
  2. The inclusion/exclusion criteria has been included as requested
  3. How the questions were posed has been included in the method section as requested.  We appreciate your valuable contributions. God Bless you